**Submission**

# Information geometry in quantum field theory: lessons from simple examples

Johanna Erdmenger[1], Kevin T. Grosvenor[1], Ro Jefferson[2]

**1** Institute for Theoretical Physics and Astrophysics and Würzburg-Dresden Cluster of Excellence ct.qmat, Julius-Maximilians-Universität Würzburg, Am Hubland, 97074 Würzburg, Germany
**2** Max Planck Institute for Gravitational Physics (Albert Einstein Institute), Am Mühlenberg 1, 14476 Potsdam-Golm, Germany

## Abstract

Motivated by the increasing connections between information theory and high-energy physics, particularly in the context of the AdS/CFT correspondence, we explore the information geometry associated to a variety of simple systems. By studying their Fisher metrics, we derive some general lessons that may have important implications for the application of information geometry in holography. We begin by demonstrating that the symmetries of the physical theory under study play a strong role in the resulting geometry, and that the appearance of an AdS metric is a relatively general feature. We then investigate what information the Fisher metric retains about the physics of the underlying theory by studying the geometry for both the classical 2d Ising model and the corresponding 1d free fermion theory, and find that the curvature diverges precisely at the phase transition on both sides. We discuss the differences that result from placing a metric on the space of theories vs. states, using the example of coherent free fermion states. We also clarify some misconceptions in the literature pertaining to different notions of flatness associated to metric and non-metric connections, with implications for how one interprets the curvature of the geometry. Our results indicate that in general, caution is needed when connecting the AdS geometry arising from certain models with the AdS/CFT correspondence, and seek to provide a useful collection of guidelines for future progress in this exciting area.

## 1 Introduction

Recent progress in understanding the AdS/CFT correspondence has seen an explosion of effort at the interface of information theory and both quantum field theory and gravity. For example, ideas such as quantum error correction appear to play a key role in bulk reconstruction, and tensor networks have become popular toy models for constructing bulk-boundary maps in this language—see, e.g., [1–4], or [5] for a recent review. Additionally, key advances in our understanding have relied crucially on entanglement-based probes of the bulk, such as Ryu-Takayanagi / HRT surfaces and their quantum extensions [6–8], which represent a fundamental link between (quantum) information-theoretic quantities on the one hand, and bulk geometric objects on the other. A further example is given by holographic distance measures that were considered both for pure states [9] and mixed states [10].

In light of the wealth of developments arising from the application of concepts from information theory to AdS/CFT, one may also go one step further, and attempt to use information theory to understand how the duality itself may arise. That is, instead of taking gauge/gravity duality as a *fait accompli* and then using information theory to fill out the holographic dictionary, one may ask whether the dual gravity theory itself can be understood as the "information space" naturally associated to the field theory. This is similar in spirit to the "It from Qubit" initiative/paradigm, in which the gravitational theory is viewed as emerging from the entanglement structure of the boundary field theory. Here, the idea is slightly broader insofar as we do not limit ourselves to entanglement-based probes, but instead ask whether there is any sense in which a geometric space can be naturally associated to the field theory based on the information content therein. For similar efforts in the context of string theory, see [11].

In fact, the study of the geometry naturally associated with a space of probability distributions is an old subject that predates AdS/CFT by several decades. Known as information geometry, it was primarily developed by statisticians, based on the pioneering work of Fisher [12]. The application of information geometry to statistical physics and thermodynamics was originally pushed by Ruppeiner [13] (see also [14]), but it enjoys a range of applications from statistics to machine learning [15–20]. The canonical reference is [21]; see also [22, 23] for some historical works, or [24] for an online review. The basic idea is to endow a statistical model with the structure of Riemannian manifold, so that methods of differential geometry can be applied to the study of probability theory and statistics. A central object in this study is the Fisher information matrix, which provides a metric on the space of distributions representing the model or theory under consideration. In the interest of making this paper self-contained, we start with a brief introduction to

information geometry in section 2.

A number of works have sought to understand AdS/CFT in this context, which requires parametrizing the theory in such a way that this statistical framework can be adapted to quantum field theory. In one approach to this task, the bulk spacetime arises as the moduli space of instantons endowed with the Fisher information metric. Building on earlier results that showed Yang-Mills instantons to be good probes of the bulk geometry [25], Blau and Thompson [26] evaluated the information metric on the moduli space of SU(2) instantons and showed that it corresponds to $AdS_5$, even away from the large $N$ limit for $\mathcal{N} = 4$ as well as $\mathcal{N} = 2$ SU($N$) super Yang-Mills theory. The instanton correction matches the first-order string theory correction to the supergravity action. An approach to capturing the compact space, the analogue of the $S^5$ in the best-known example of AdS/CFT, was taken more recently in [27] using the $\mathbb{CP}^N$ nonlinear sigma model as a proxy; see also [28–31] for related works. Yet a different approach was used in [32] to construct a metric in real Euclidean space by considering an RG gradient flow for large $N$ $\phi^4$ theory in $d$ dimensions. A ($d+1$)-dimensional asymptotically AdS metric was found both in the UV and in the IR, with different AdS radii. To date, it is unclear how to capture the internal $S^5$ in such an approach, but the appearance of an AdS space has nonetheless generated some excitement.

However, a key ingredient in the above is the fact that in the large $N$ saddle point approximation, the classical supergravity action takes on a Gaussian form. Hence, while we do not know the degrees of freedom in the strongly coupled $\mathcal{N} = 4$ gauge theory, the duality

$$\left\langle e^{\int \mathrm{d}^d x \phi_0(\mathbf{x}) \mathcal{O}(\mathbf{x})} \right\rangle_{\mathrm{CFT}} = e^{-S_{\mathrm{SUGRA}}} \bigg|_{\phi(0,\mathbf{x})=\phi_0(\mathbf{x})} , \tag{1}$$

implies that they lead to a Gaussian probability distribution. But it has long been known in the information geometry literature, even prior to AdS/CFT, that the information metric associated with Gaussian distributions is hyperbolic space; see section 2 and [21, 33] for more details. In fact, as we shall show in section 3, the AdS metric appears to be merely a reflection of the symmetries of the underlying distribution. This may have important implications in light of recent attempts to connect the appearance of an AdS geometry with holography [34–38]. Furthermore, the map between Gaussian distributions and a hyperbolic Fisher metric is not bijective: other, quite different distributions may also lead to the same hyperbolic space; see section 3 and [39], in which a construction procedure for obtaining probability distributions from a fixed information metric was given.

These two issues – symmetry and non-uniqueness – immediately raise two questions: first, how much physics of the field theory is encoded in the information geometry? Second, can we find other meaningful realizations of the information space associated to a given theory, and perhaps even generalize gauge/gravity duality to an "information/geometry" duality applicable to a wider class of theories? The purpose of this work is to present some initial explorations into these questions, as well as to collect some facts about the Fisher metric which, while familiar to experts in information geometry, do not appear to have survived the latter's recruitment into the quantum field theory community.

As mentioned above, we shall begin by reviewing the basic ingredients of information geometry in section 2, and show how AdS appears as the metric on the space of Gaussians as an illustrative example. In section 3, we substantiate the aforementioned claim that the hyperbolic geometry is a reflection of the symmetries of the underlying theory, and discuss some further issues with non-uniqueness in the putative information $\leftrightarrow$ geometry map. This motivates us to consider which physical features are faithfully preserved under this relationship. To that end, we present in section 4 an example of an unstable system – massless $\phi^4$ theory in 3+1 dimensions with an inverted potential – and show that the

information geometry on the space of instantons is complex, and therefore not Euclidean AdS. We interpret this as the geometrical manifestation of the sickness of the underlying field theory.

We then turn to the study of curvature invariants. Insofar as these are fundamental features of the geometry, it is natural to ask which information-theoretic aspects may be encoded therein. Motivated in part by earlier work in condensed matter systems [40–42] (see also [43]), we shall examine the 2d classical Ising model in section 5, and show that the Ricci scalar diverges along the critical line. Additionally, using the well-known map between the 2d classical Ising model and a 1d free fermion theory, this example provides us with the opportunity to examine the information geometry on both sides of an existing correspondence between physical theories. We shall find that the geometry of the free fermion theory is one-dimensional: the single component of the metric can be parametrized by the fermion mass, and diverges precisely at the critical temperature in a way that matches the behaviour of the 2d theory. Thus, while the correspondence between the 2d classical Ising model and the 1d free fermion theory is not a true duality, the salient geometrical features are captured on both sides. To our knowledge, this is the first such application of information geometry to both sides of an existing correspondence between different physical theories, and it would be very interesting to consider other examples.

We note that an important difference arises when considering the Fisher metric on the space of theories, spanned by its couplings and masses, as opposed to the metric on the space of states. The former was considered for quantum field theories for instance in [44], where the Zamolodchikov metric is used as an information metric, which then changes under RG flow. The latter was considered for instance in the instanton approach of Blau and Thompson already discussed above [26], where the information metric is defined on the moduli space of the gauge theory considered (see also [45]). Our treatment of the Ising model in section 5 falls into the former class. To illustrate the difference however, we also discuss the information metric on the space of coherent states for free fermions in section 6.

There has been some confusion in the physics literature as to the curvature of the information space. For example, [41] asserts that the geometry of non-interacting models is flat, while this is clearly false for even the simple Gaussian example mentioned above. We believe this is a confusion of language, stemming from the fact that in information geometry, one typically considers the 1-connection, rather than the 0-connection familiar to physicists; and the associated 1-curvature is indeed zero for a wide class of models, known as exponential families (see sections 2.1). The reason for this stems from the fact that the 1-curvature is more naturally associated with information loss along a curve, as quantified by the Kullback-Leibler divergence or relative entropy, whereas the information-theoretic interpretation of the 0-curvature is less clear. A skim of the information geometry literature will hence turn up statements about flat geometries, without making it immediately obvious that this is not flatness in the physicist's familiar sense. We elaborate on this point in section 7, where we also point out that vanishing 1-curvature corresponds to the trivial solution of the equation of motion arising from a Chern-Simons action. This is reminiscent of the map between field theory and supergravity actions in standard realizations of field theory/gravity dualities, at least insofar that there is an obvious action providing the required dynamics on the gravity side.

Finally, we close in section 8 with a concise summary of the lessons learned from the examples herein, as well as some speculations on the relationship between information geometry and holographic RG [44], and the potential for this approach to enable us to compute complexity [46] in strongly coupled / interacting field theories.

## 2 Information geometry

To make this paper self-contained, let us begin with a brief introduction to information geometry. We shall present only those ingredients required for subsequent sections, and refer the interested reader to [21] for details.

One is interested in studying the properties of a *statistical model* $S$, which is essentially a set of probability distribution functions $p : \mathcal{X} \to \mathbb{R}$ satisfying [1]

$$p(x) \geq 0 \quad \forall x \in \mathcal{X} \;, \quad \text{and} \quad \int \mathrm{d}x \, p(x) = 1 \;, \tag{2}$$

where $\mathcal{X}$ is the space of stochastic or physical variables (e.g., $\mathbb{R}^n$, or some discrete set). Additionally, each $p$ may be parametrized by $\xi = (\xi^1, \dots, \xi^n) \in \mathbb{R}^n$, so that the model $S$ is

$$S = \{ p_\xi = p(x; \xi) \,|\, \xi \in \Xi \} \;, \tag{3}$$

where $\Xi \subset \mathbb{R}^n$, and $\xi \mapsto p_\xi$ is injective and provides a map between the parameter space and the points on the manifold. That is, we regard each point $\xi$ as a different distribution within our model, and take the map $\Xi \to \mathbb{R}$ provided by $\xi$ to be in $C^\infty$ so that we may take derivatives with respect to these parameters. Note that it is the parameters $\xi$ – which represent points on this statistical manifold – with respect to which we will be computing derivatives, not the stochastic variables $x$. Accordingly, we denote $\partial_i \equiv \frac{\partial}{\partial \xi^i}$.

Now, given a model $S$, the *Fisher information matrix* of $S$ at a point $\xi$ is the $n \times n$ matrix $G(\xi) = [g_{ij}(\xi)]$, with elements

$$g_{ij}(\xi) \equiv \langle \partial_i \ell_\xi \partial_j \ell_\xi \rangle_\xi = \int \mathrm{d}x \, \partial_i \ell(x; \xi) \partial_j \ell(x; \xi) p(x; \xi) \;, \tag{4}$$

where

$$\ell_\xi(x) = \ell(x; \xi) \equiv \ln p(x; \xi) \;, \tag{5}$$

and the expectation $\langle \dots \rangle_\xi$ with respect to the distribution $p_\xi$ is defined as

$$\langle f \rangle_\xi \equiv \int \mathrm{d}x \, f(x) p(x; \xi) \;. \tag{6}$$

Note that $G$ is symmetric and positive semi-definite by construction. It will prove convenient to rewrite the metric as

$$
\begin{aligned}
g_{ij} = \langle \partial_i \ell \partial_j \ell \rangle &= \int \mathrm{d}x \, p \, \partial_i \ln p \, \partial_j \ln p = \int \mathrm{d}x \, \partial_i p \, \partial_j \ln p \\
&= \int \mathrm{d}x \, \partial_i (p \, \partial_j \ln p) - \int \mathrm{d}x \, p \, \partial_i \partial_j \ln p \\
&= \partial_i \langle \partial_j \ell \rangle - \langle \partial_i \partial_j \ell \rangle \;,
\end{aligned}
\tag{7}
$$

where on the last line, the first term vanishes by the normalization constraint, i.e., $\langle \partial_i \ell \rangle = \partial_i \langle 1 \rangle = 0$.

### 2.1 Exponential families

While the machinery of information geometry can be applied to any distribution which satisfies the condition (2), we will be particularly interested in a class of models known as

---

[1]More generally, this framework goes through provided the distributions are normalized to some finite value, not necessarily 1.

*exponential families.* Suppose an $n$-dimensional model $S = \{p_\theta \,|\, \theta \in \Theta\}$ can be expressed in terms of $n+1$ functions $\{C, F_1, \ldots, F_n\}$ on $\mathcal{X}$ and a function $\psi$ on $\Theta$ as

$$p(x; \theta) = \exp\left[C(x) + \theta^i F_i(x) - \psi(\theta)\right] \ , \tag{8}$$

where Einstein's summation convention is assumed. Then $S$ is an exponential family, and $\theta$ are the so-called *canonical coordinates*, not to be confused with the stochastic variable $x$. The function $\psi$ is known as the potential, which is fixed by the normalization to

$$\psi(\theta) = \ln \int \mathrm{d}x \exp\left[C(x) + \theta^i F_i(x)\right] \ . \tag{9}$$

Note that the parametrization $\theta \mapsto p_\theta$ is 1:1 if and only if the functions $\{C, F_1, \ldots, F_n\}$ are linearly independent, which we shall assume to be the case.

Many important models fall into this class which, in addition to some interesting mathematical properties (see sec. 7), have the technical nicety of admitting an expression for the Fisher metric directly in terms of the potential. Observe that

$$\partial_j \ell = F_j(x) - \partial_j \psi(\theta) \quad \Longrightarrow \quad \partial_i \partial_j \ell = -\partial_i \partial_j \psi(\theta) \ , \tag{10}$$

where the derivatives are taken with respect to the canonical coordinates, $\partial_i \equiv \frac{\partial}{\partial \theta^i}$. Therefore the metric (7) may be written

$$g_{ij} = \partial_i \partial_j \psi(\theta) \ . \tag{11}$$

We emphasize that while (7) is true for general models, the simple form (11) holds only for exponential families. We will rely on this form of the metric extensively below. Another form that is sometimes useful is the expression in terms of the covariance matrix of $F_i$,

$$g_{ij} = \langle (F_i - \langle F_i \rangle)(F_j - \langle F_j \rangle) \rangle \ . \tag{12}$$

## 2.2 Simple example: AdS$_2$ from a Gaussian

As a concrete example, let us show how hyperbolic space arises from the normal distribution

$$p(x; \mu, \sigma) = \frac{1}{\sqrt{2\pi}\sigma} e^{-\frac{(x-\mu)^2}{2\sigma^2}} = \exp\left[-\frac{1}{2\sigma^2}\left(x^2 - 2\mu x + \mu^2\right) - \ln(\sqrt{2\pi}\sigma)\right] \ . \tag{13}$$

This clearly falls within the class of exponential families (8), with the identifications:[2]

$$C(x) = 0 \ , \quad F_1(x) = x \ , \quad F_2(x) = x^2 \ , \quad \theta_1 = \frac{\mu}{\sigma^2} \ , \quad \theta_2 = -\frac{1}{2\sigma^2} \ ,$$
$$\psi(\mu, \sigma) = \frac{\mu^2}{2\sigma^2} + \ln\left(\sqrt{2\pi}\sigma\right) \ . \tag{14}$$

To evaluate the metric (11), we invert these relations in order to express the distribution in terms of the canonical coordinates:

$$\mu = -\frac{\theta_1}{2\theta_2} \ , \qquad \sigma = \frac{1}{\sqrt{-2\theta_2}} \ , \qquad \psi(\theta) = -\frac{\theta_1^2}{4\theta_2} + \frac{1}{2}\ln\left(-\frac{\pi}{\theta_2}\right) \ . \tag{15}$$

Strictly speaking, the potential is all we need to compute the metric, but we can also write out the distribution (13) as a quick check on the consistency of our identifications:

$$p(x; \theta) = \sqrt{-\frac{\theta_2}{\pi}} e^{\theta_2\left(x + \frac{\theta_1}{2\theta_2}\right)^2} = \exp\left[\theta_2 x^2 + \theta_1 x + \frac{\theta_1^2}{4\theta_2} - \frac{1}{2}\ln\left(-\frac{\pi}{\sigma_2}\right)\right] \ . \tag{16}$$

---

[2]To avoid conflating indices with powers, we have lowered the indices on the parameters $\theta^i = \theta_i$.

Substituting the potential $\psi(\theta)$ in (15) into (11) then gives the metric in canonical coordinates:

$$g_{ij} = -\frac{1}{2\theta_2}\begin{pmatrix} -1 & \frac{\theta_1}{\theta_2} \\ \frac{\theta_1}{\theta_2} & \frac{\theta_2 - \theta_1^2}{\theta_2^2} \end{pmatrix} , \tag{17}$$

and hence the squared line element is

$$\mathrm{d}s^2 = -\frac{1}{2\theta_2}\mathrm{d}\theta_1^2 + \frac{\theta_2 - \theta_1^2}{2\theta_2^3}\mathrm{d}\theta_2^2 + \frac{\theta_1}{\theta_2^2}\mathrm{d}\theta_1\mathrm{d}\theta_2 . \tag{18}$$

The metric in terms of the original (physical) coordinates $\mu, \sigma$ is then obtained by performing a simple change of basis via the identifications (14):

$$\mathrm{d}\theta_i = \frac{\partial\theta_i}{\partial\mu}\mathrm{d}\mu + \frac{\partial\theta_i}{\partial\sigma}\mathrm{d}\sigma \quad\Longrightarrow\quad \begin{cases} \mathrm{d}\theta_1 = \frac{1}{\sigma^2}\mathrm{d}\mu - \frac{2\mu}{\sigma^3}\mathrm{d}\sigma \\ \mathrm{d}\theta_2 = \frac{1}{\sigma^3}\mathrm{d}\sigma . \end{cases} \tag{19}$$

whence we at last obtain

$$\mathrm{d}s^2 = \frac{\mathrm{d}\mu^2 + 2\mathrm{d}\sigma^2}{\sigma^2} , \tag{20}$$

which is none other than $\mathrm{AdS}_2$, with the standard deviation playing the role of the radial coordinate. Loosely speaking, the intuition is that a large standard deviation implies a large overlap between different distributions. Operationally, this means that they are harder to distinguish (e.g., requiring more measurements), and are accordingly considered to be "close" in an information-theoretic sense.

Two immediate observations are worth remarking upon. First, we see that the geometry corresponding to a free theory is clearly not flat, contrary to some claims in the literature that non-interacting theories have vanishing Ricci scalar [41,47]; we shall return to this issue when we discuss connections in section 7. Second, we note that the map between the Gaussian and the hyperbolic Fisher metric is achieved without any reference to the dynamics leading to this particular metric on the gravity side, and in this sense falls short of the original AdS/CFT correspondence. We shall turn to this second issue in the next section, and investigate the extent to which this geometry merely reflects the underlying symmetries of the distribution.

Before doing so however, let us mention a simple example of a distribution which is *not* an exponential family and yet still yields and $\mathrm{AdS}_2$ metric, namely the Cauchy-Lorentz distribution. For other examples of distributions that yield AdS, as well as other geometries (e.g., the sphere), see [33,39].

# 3   A hyperbolic red herring[3]

When we say that the information geometry merely reflects the underlying symmetries of the distribution, we mean that a symmetry of the probability distribution will manifest itself as a corresponding symmetry of the Fisher metric. Let us be precise about what is meant by a symmetry of the probability distribution. Consider a map

$$\tilde{\xi} : \Xi \to \Xi . \tag{21}$$

Suppose that there exists a map

$$\tilde{x} : \mathcal{X} \to \mathcal{X} \tag{22}$$

---

[3]For those unfamiliar with this idiom, a red herring is a misleading or distracting clue that may cause one to reach a false conclusion.

such that

$$p(x; \tilde{\xi}) \, \mathrm{d}x = p(\tilde{x}; \xi) \, \mathrm{d}\tilde{x}. \tag{23}$$

Then, the probability distribution $p(x; \xi)$ is said to be symmetric under the transformation $\xi \mapsto \tilde{\xi}(\xi)$. In other words, by a suitable redefinition of the stochastic variable $x$, we can "undo" the transformation on $\xi$, thereby putting the transformed probability distribution back into the form of the original probability distribution.

Let us take the Gaussian distribution written in terms of the mean $\mu$ and standard deviation $\sigma$ as an example. It is clear that the translation $(\mu, \sigma) \mapsto (\mu + c, \sigma)$, where $c$ is any real constant, can be "undone" by a translation $x \mapsto x + c$. Therefore, we say that the Gaussian distribution is symmetric or invariant under a translation of the mean. A more nontrivial transformation is the scaling transformation $(\mu, \sigma) \mapsto (\lambda \mu, \lambda \sigma)$, for some real $\lambda$. The map $x \mapsto \lambda x$ will undo this scaling transformation. Unlike the translation, the scaling transformation does come with a nontrivial Jacobian in $x$, but this Jacobian is precisely what is needed to maintain the relation (23). Therefore, the Gaussian is invariant under translations of $\mu$ and simultaneous scaling of $\mu$ and $\sigma$. The Fisher metric corresponding to the family of Gaussian distributions inherits these symmetries. But the only metric which enjoys these symmetries is $\mathrm{AdS}_2$ and therefore the Fisher metric could not have been anything else!

To reiterate, a symmetry of the probability distribution will also be a symmetry of the corresponding Fisher metric. The converse, however, is not necessarily true—the Fisher metric can exhibit *more* symmetries than are present in any particular probability distribution from which that metric may be derived. Note that we are careful to say "any particular probability distribution" because, as demonstrated in [39], there are in fact infinitely many probability distributions that give the same Fisher metric. For instance, example 4.8 of [39] derives Euclidean $\mathrm{AdS}_2$ as the Fisher metric of the following three-dimensional probability distribution (in this case, $\mathcal{X} = \mathbb{R}^3$):

$$p(x_1, x_2, x_3; \mu, \sigma) = \prod_{i=1}^{3} p_i(x_i - h_i), \tag{24}$$

where

$$p_1(x) = \frac{1}{\sqrt{2\pi}} e^{-\frac{1}{2}x^2} \, , \qquad h_1(\mu, \sigma) = \frac{\cos \mu}{\sigma} \, ,$$

$$p_2(x) = \frac{1}{\pi} \operatorname{sech} x \, , \qquad h_2(\mu, \sigma) = \frac{\sin \mu}{\beta} \, , \tag{25}$$

$$p_3(x) = \frac{1}{\pi(1 + x^2)} \, , \qquad h_3(\mu, \sigma) = \ln\big(\sigma + \sqrt{\sigma^2 - 1}\big) - \frac{\sqrt{\sigma^2 - 1}}{\sigma} \, .$$

It is clear that this probability distribution exhibits neither the translation invariance in $\mu$ nor the scaling symmetry in $\mu$ and $\sigma$ that is enjoyed by the corresponding Fisher metric, which is Euclidean $\mathrm{AdS}_2$.

Given the above, we believe that care is needed when attempting to identify the appearance of an AdS geometry with some intrinsic aspect of holography. This is not to say that the hyperbolic geometry arising from some theories is unrelated to AdS/CFT, but it is clearly not unique, and may represent a red herring or false conclusion in this context.

## 4  Unstable configurations

Here we turn to a simple example demonstrating how the fact that a system has an unstable potential is encoded in the Fisher metric. Our example is constructed on the

space of states, in analogy with the stable examples for the Fisher metric of Yang-Mills instantons in [26] and the Klein-Gordon field in [48]. Both of these make use of the proposal of Hitchin [49] to identify the negative of the Lagrangian density, evaluated on a family of field configurations, as the probability distribution on those states. The Fisher metric on the space of 4D Yang-Mills instantons turns out to be Euclidean AdS$_5$ [26]. In that case, the Lagrangian density $F^2$ evaluated on the instantons exhibits translation invariance in the center and scale invariance in both the center and the width of the instantons.

Our example is similar to the Yang-Mills instanton, but with one crucial difference – that the instantons themselves are in an unstable potential. Of course, the same symmetry argument would make one conclude that the Fisher metric ought to still be AdS. However, we will see how the machinery is smart enough to know when the field theory that one is considering is unstable. The system is taken to be in four Euclidean dimensions with a real scalar field and an action

$$S = \int \mathrm{d}^4 x \left( \partial_\mu \phi \, \partial^\mu \phi - g^2 \phi^4 \right). \tag{26}$$

In Euclidean signature, the potential is $V = -g^2 \phi^4$, which is unbounded from below. Therefore, at least classically, the theory is unstable. Nevertheless, one can demonstrate the existence of exact solutions of the equation of motion given by the instantons, parametrized by a four-dimensional center, $\xi^\mu$ and a width $\rho$:

$$\phi(x; \xi^\mu, \rho) = \frac{2\rho}{g\left[ (x - \xi)^2 + \rho^2 \right]}. \tag{27}$$

The action evaluated on this instanton can be normalized to one by setting

$$g^2 = \frac{8\pi^2}{3}. \tag{28}$$

The resulting integrals that appear in the Fisher metric are technically ill-defined since the integrands contain singularities. For example, the non-vanishing components of the Fisher metric read

$$g_{\rho\rho} = \frac{28a}{5\rho^2} + \frac{24a}{\rho^2} \int_0^\infty \frac{x\,\mathrm{d}x}{(x-1)(x+1)^4}, \tag{29a}$$

$$g_{\xi^\mu \xi^\nu} = \left( \frac{28a}{5\rho^2} + \frac{6a}{\rho^2} \int_0^\infty \frac{x^2 \mathrm{d}x}{(x-1)(x+1)^4} \right) \delta_{\mu\nu}, \tag{29b}$$

where $a$ is an overall prefactor that we can set at our convenience. The remaining integrals have a pole at $x = 1$ and thus the integrals themselves are ill-defined: one must specify a contour of integration and thus a way of avoiding the pole. There are two ways of doing this: Either add or subtract $i\epsilon$ to $x$ and then take the $\epsilon \to 0$ limit at the end. The results for the Fisher metric components are denoted with a superscript $\pm$ depending on which sign $\pm i\epsilon$ prescription is used,

$$g_{\rho\rho}^\pm = \frac{36 \mp 15i\pi}{10} \frac{a}{\rho^2}, \tag{30a}$$

$$g_{\xi^\mu \xi^\nu}^\pm = \frac{244 \mp 15i\pi}{40} \frac{a\delta_{\mu\nu}}{\rho^2}. \tag{30b}$$

For convenience, we may set $a$ such as to set $g_{\xi^\mu \xi^\nu}$ equal to $\frac{1}{\rho^2}\delta_{\mu\nu}$. Then, when the dust settles, the squared line element may be expressed in the form

$$\mathrm{d}s^2 = \frac{\mathrm{d}\xi_\mu \mathrm{d}\xi^\mu + c\,\mathrm{d}\rho^2}{\rho^2}, \tag{31}$$

where $c$ is some fixed complex number. Importantly, $c$ has a nonzero imaginary part and therefore, this metric cannot be considered real Euclidean AdS$_5$. This signals the instability present in the field-theory model, and indicates that the information geometry retains knowledge of whether the original field theory is well-defined.

# 5 Information geometry on theory space: the Ising model

Here we perform a further investigation of which aspects of the physical theory are captured by the information geometry by considering the 2d Ising model. Studies of aspects of information geometry for the Ising and related spin models have been performed before in the literature (see, e.g., [41, 43]). Here we focus on the 2d Ising model, which has the feature of admitting a map to an ostensibly quite different physical theory, namely a 1d free fermion field theory. In the first subsection, we shall examine the Fisher metric and its Ricci curvature for the 2d Ising model on the theory space spanned by its two couplings, and show that the geometry correctly captures the divergence along the critical line. We will then reproduce this behaviour in the second subsection for the 1d free fermion theory.

## 5.1 2d classical Ising model

Let us consider the 2d classical Ising model on a square lattice of spins $\sigma_{i,j} = \pm 1$, with vanishing external magnetic field. Denoting the horizontal and vertical couplings by $J$ and $K$, respectively, the Hamiltonian for an $N \times N$ lattice may be written

$$H = -J \sum_{i,j=1}^{N} \sigma_{i,j}\sigma_{i+1,j} - K \sum_{i,j=1}^{N} \sigma_{i,j}\sigma_{i,j+1} \; , \tag{32}$$

where we have identified both directions to form a torus, i.e, $\sigma_{N+1,j} = \sigma_{1,j}$ and $\sigma_{i,N+1} = \sigma_{i,1}$. Note that the state (i.e., spin configuration) satisfies a Boltzmann distribution at inverse temperature $\beta$, and thus we may write

$$p(\sigma) = Z^{-1}e^{-\beta H(\sigma)} \; , \tag{33}$$

where the partition function is

$$Z = \prod_i \sum_{\sigma_i = \pm 1} e^{-\beta H(\sigma)} \; . \tag{34}$$

Hence, by exponentiating this normalization factor, we may express the distribution in the form of an exponential family:

$$p(\sigma;\theta) = \exp\left\{ \beta \left[ J \sum_{i,j=1}^{N} \sigma_{i,j}\sigma_{i+1,j} + K \sum_{i,j=1}^{N} \sigma_{i,j}\sigma_{i,j+1} \right] - \ln Z \right\} \; , \tag{35}$$

cf. (8) where the canonical coordinates $\theta \in \{\beta J, \beta K\}$. Note that for distributions in the form (33), $\ln Z$ plays the role of the potential $\psi$, which means that in order to compute the metric (11), all we need is an expression for the free energy. In particular, in thermodynamic limit $N \to \infty$, the reduced free energy per site $f = -\frac{\beta}{N}F = N^{-1}\ln Z$ becomes

$$f = \frac{1}{2}\ln 2 + \frac{1}{2\pi}\int_0^{\pi} d\phi \, \ln\left[ \cosh(2\beta J)\cosh(2\beta K) + \frac{1}{k}\sqrt{1 + k^2 - 2k\cos(2\phi)} \right] \; , \tag{36}$$

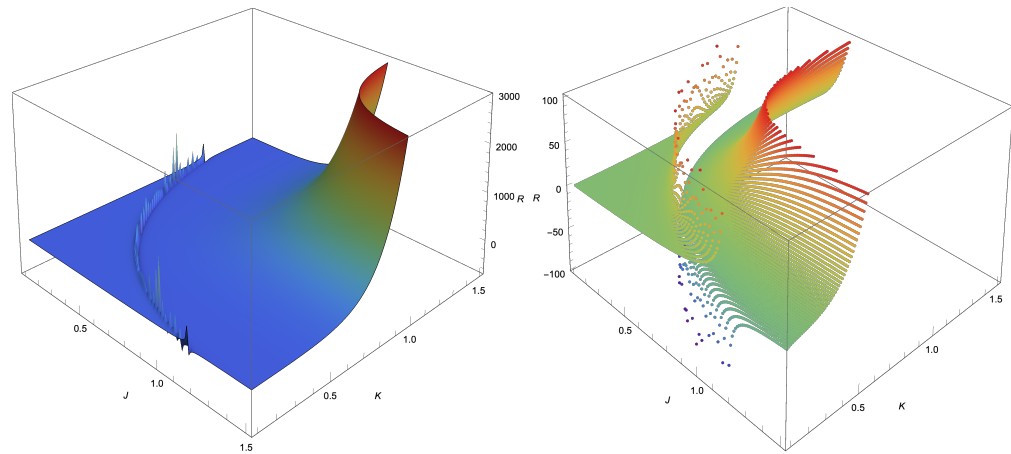

Figure 1: Numerical evaluation of the Ricci scalar (39) for the 2d Ising model, using a uniform grid of $141 \times 141 = 19,881$ points between $J, K \in \{0.1, 1.5\}$ (with $\beta = 1$). In the left figure, we have interpolated between points to better show the global features of the curvature: it diverges as $J, K \to \infty$, has a discontinuity along the critical line (40) (rendered as the jagged line of spikes by Mathematica's interpolation attempts), and is otherwise approximately flat. In the right figure, we have plotted the data without interpolating, as well as restricted the vertical range to $R \in \{-100, 100\}$, to better show the discontinuity along the critical line (40). One sees clearly that the curvature diverges in opposite directions depending on which side one approaches from. We shall see below that this corresponds to the sign of the mass in the corresponding free fermion theory on either side of the phase transition, cf. fig. 2.

where we have defined $k \equiv \operatorname{csch}(2\beta J) \operatorname{csch}(2\beta J)$, and $\phi$ is some auxilliary angular parameter. This will play the role of the potential, i.e.,

$$g_{ij} = \partial_i \partial_j f \ . \tag{37}$$

The integral over the auxilliary parameter $\phi$ cannot be performed analytically, but we can nonetheless proceed to determine the form of the metric, and then evaluate the curvature numerically. To compute the derivatives in (37), we first re-express the free energy in terms of the canonical coordinates. By inspection, we identify

$$C = 0 \ , \quad F_1 = \sigma_{i,j}\sigma_{i+1,j} \ , \quad F_2 = \sigma_{i,j}\sigma_{i,j+1} \ , \quad \theta_1 = \beta J \ , \quad \theta_2 = \beta K \ , \quad \psi(\theta) = \ln Z \ . \tag{38}$$

Note that $F_1$ and $F_2$ are linearly independent, as required, since they correspond to couplings along different axes. In practice, we will be interested in evaluating the curvature for a range of couplings at fixed temperature, so we may equivalently absorb $\beta$ into the couplings (i.e., $\beta = 1$), whereupon the canonical and physical variables coincide.

The metric (37) is rather unwieldy, so we do not write out the full expression here. However, we can still proceed to compute curvature invariants – in particular the Ricci scalar – to see which physical aspects of the 2d Ising model are reflected in the geometry. To evaluate the Ricci scalar, it is convenient to use the following expression in terms of the reduced free energy [41]:

$$R = -\frac{1}{2g^2} \begin{vmatrix} \partial_i^2 f & \partial_i \partial_j f & \partial_j^2 f \\ \partial_i^3 f & \partial_i^2 \partial_j f & \partial_i \partial_j^2 f \\ \partial_i^2 \partial_j f & \partial_i \partial_j^2 f & \partial_j^3 f \end{vmatrix} \ , \tag{39}$$

where $g = \det g_{ij}$. This provides a (very lengthy) expression for the Ricci curvature, which can be evaluated numerically for a range of couplings $J, K$. Results are shown in figure 1.

Of course, one of the most important features of the 2d Ising model is the presence of a finite-temperature phase transition along the critical line

$$\sinh(2\beta J)\sinh(2\beta K) = 1 \, , \tag{40}$$

which manifests in the information geometry as the discontinuity in the curvature observed in fig. 1. It is encouraging that this important physical feature is captured by this approach, though it remains an open question as to which other aspects of the geometry are faithful representations of the original physics. For example, in the context of our earlier remark that non-interacting theories do *not* necessarily imply $R=0$, here we have an example of the converse, namely that the vanishing of the Ricci scalar away from the critical line (and the divergence near infinity) clearly does *not* imply that the model ceases to be interacting. Thus, while the curvature clearly does capture important physical features (e.g., critical points), a complete mapping between the physics of the underlying model and the curvature of the geometry requires a more careful study. Indeed, as we discuss in section 7, the notion of curvature studied above is not necessarily the most appropriate for capturing certain information-theoretic notions.

## 5.2   1d free fermion theory

It is a well-known fact that the 2d classical Ising model can be mapped to a theory of non-interacting fermions in 1d. There are however myriad different ways of actually constructing the resulting field theory which, while they reproduce the correct critical behaviour, give slightly different expressions for the fermion mass in terms of the original 2d couplings; for a selection of different results [50–53]. Since we are primarily interested in comparing the critical behaviour, we shall proceed with [50], which – in the notation above – corresponds to setting $J = K = 1$, i.e., we treat $\beta$ as the coupling and examine the geometry along the line of $J \leftrightarrow K$ symmetry in fig. 1. Accordingly, we expect that the information geometry for the 1d free fermion theory will diverge at the critical (inverse) temperature

$$\beta_c = \frac{1}{2}\ln\bigl(\sqrt{2}+1\bigr) \approx 0.440687 \, , \tag{41}$$

cf. (40) with $J = K = 1$ and $\beta = \beta_c$. We shall now see that the information geometry indeed reproduces this feature.

For the isotropic case with $J=K$ absorbed into $\beta$, we shall write the partition function (34) as

$$Z_V(\beta) = \frac{1}{2^V} \sum_{\sigma_i=\pm 1} \exp\Bigl(\beta \sum_{(ij)} \sigma_i\sigma_j\Bigr). \tag{42}$$

where the sum in the exponential is over nearest-neighbor pairs $(ij)$ and where we have included an explicit normalization by the total Hilbert space dimension $2^V$, where $V = N^2$ is the total number of sites, for consistency with [50]. In this notation, the corresponding reduced free energy is[4]

$$f = \lim_{V\to\infty} \frac{1}{V}\ln Z_V(\beta). \tag{43}$$

In this case, we have only a single parameter $\beta$, so the information metric is one-dimensional, cf. (37) with $i=j=\beta$. The second derivative of the reduced free energy with respect to $\beta$

---

[4]Note that [50] writes this as $F(\beta)$, as if this were the free energy. This is simply a matter of nomenclature. However, the standard definition of the free energy is $F = -\frac{1}{\beta}\ln Z$, which explains why we put a relative factor of $-\beta$ between $F$ and the reduced free energy $f$. At the end of the day, we want to take two derivatives of $\ln Z$ to get the metric. Dividing by the volume before taking derivatives ensures a well-defined continuum limit $V \to \infty$.

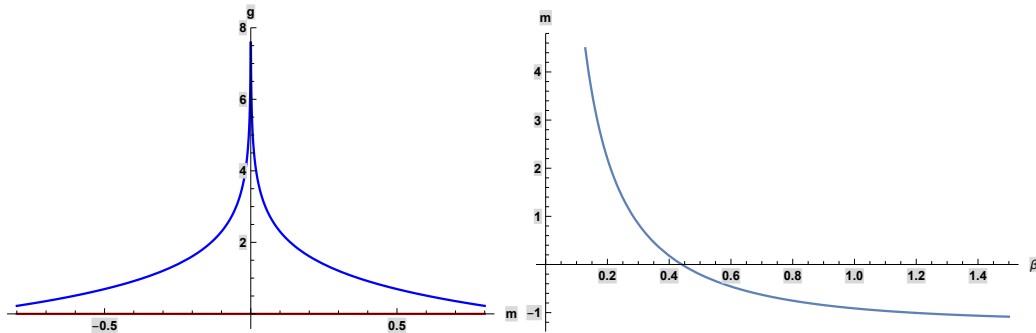

Figure 2: Left: real (blue) and imaginary (red) components of the Fisher information metric (44) for the 1d free fermion theory corresponding to the 2d classical Ising model. Note that the metric obtained via the mapping in [50] is only valid near the critical point $m_c = 0$. Right: plot of the mass (46) as a function of $\beta = \beta J = \beta K$, from which we see that the free fermion mass vanishes precisely at the critical temperature (41), corresponding to the phase transition in the 2d model where one expects to find a conformal field theory. The mass also takes opposite signs on either side of the critical line in fig. 1 that matches the direction of divergence in the 2d curvature, i.e., we find $m > 0$ on the side nearer the origin and $m < 0$ on the side nearer $\beta J = \beta K \to \infty$.

is essentially the specific heat, which is given in [50],

$$g_{\beta\beta} = \frac{\mathrm{d}^2 F(\beta)}{\mathrm{d}\beta^2} \simeq \frac{8\sqrt{2}}{\pi} \ln \frac{1}{|\beta - \beta_c|} + \text{regular part}. \tag{44}$$

The regular part consists of terms which are polynomial in $\beta - \beta_c$ and thus vanish as $\beta \to \beta_c$; since we are interested in the dominant behaviour near criticality, we shall discard this piece henceforth. We will also disregard the overall prefactor, since this does not alter the divergence structure of the metric. Thus, for our purposes, we may summarize the result more compactly as

$$g_{\beta\beta} \approx \ln \frac{1}{|\beta - \beta_c|}, \tag{45}$$

which diverges to $+\infty$ as $\beta \to \beta_c$.

We wish to express (44) in terms of the mass parameter $m$ that appears in the corresponding free Majorana fermion field theory. This model is also described in [50], with

$$m = 2\left(\frac{t_c}{t} - 1\right), \tag{46}$$

where $t_c \equiv \sqrt{2} - 1$ and $t \equiv \tanh\beta$ (the lattice spacing has been set to 1, so $m$ is dimensionless). Solving this expression for $\beta$ perturbatively around the critical point yields

$$\beta - \beta_c \approx -\frac{m}{4}, \tag{47}$$

which we can then substitute into the metric component (45) to find

$$g_{\beta\beta} \approx \ln \frac{1}{|m|}, \tag{48}$$

where we have again dropped the regular piece and overall numerical prefactor. This result is plotted in fig. 2.

We would now like to reproduce this result from a calculation in the low-energy effective field theory whose action is given, in Euclidean signature, by

$$S = \int \frac{\mathrm{d}^2 z}{2\pi} \left( \psi \bar{\partial} \psi + \overline{\psi} \partial \overline{\psi} + i m \overline{\psi} \psi \right). \tag{49}$$

The lightcone coordinates $(z, \bar{z})$ are related to the Cartesian coordinates $(x, y)$ via

$$\begin{pmatrix} z \\ \bar{z} \end{pmatrix} = \begin{pmatrix} 1 & i \\ 1 & -i \end{pmatrix} \begin{pmatrix} x \\ y \end{pmatrix} , \qquad\qquad \begin{pmatrix} x \\ y \end{pmatrix} = \frac{1}{2} \begin{pmatrix} 1 & 1 \\ -i & i \end{pmatrix} \begin{pmatrix} z \\ \bar{z} \end{pmatrix} , \qquad (50)$$

and the lightcone derivatives are defined as

$$\partial \equiv \partial_z , \qquad\qquad \bar{\partial} \equiv \partial_{\bar{z}} . \qquad (51)$$

To calculate the metric $g_{\beta\beta}$ for this theory, we first note that in the field theory analogue of the probability distribution (33), one identifies the probability of a particular field configuration with the integrand of the normalized path integral (in Euclidean signature) evaluated on that field configuration. One can then show (cf. eqs. (11) and (12) in [54]) that the Fisher metric is given by

$$g_{ij} = \frac{1}{V} \Big( \langle \partial_i S \, \partial_j S \rangle - \langle \partial_i S \rangle \langle \partial_j S \rangle \Big) , \qquad (52)$$

where the expectation values are taken with respect to the standard path integral, the derivatives are taken with respect to the coupling constants of the theory, and the volume divergence is explicitly divided out as usual. In the particular case when the coupling constant is a mass parameter, the first expectation value above reduces to a four-point function and the second to a product of two-point functions. In the special case of a free field theory, of which (49) is an example, the one can use Wick's theorem to reduce the four-point function to sums (or differences) of products of pairs of two-point functions. For the example at hand, one finds

$$g_{mm} = \int \mathrm{d}^2 x \Big( \langle \overline{\psi}(x) \, \overline{\psi}(0) \rangle \langle \psi(x) \, \psi(0) \rangle - \langle \overline{\psi}(x) \, \psi(0) \rangle \langle \psi(x) \, \overline{\psi}(0) \rangle \Big) . \qquad (53)$$

The set of four possible two-point functions is given by [50]

$$\begin{pmatrix} \langle \psi_1 \psi_2 \rangle & \langle \psi_1 \overline{\psi}_2 \rangle \\ \langle \overline{\psi}_1 \psi_2 \rangle & \langle \overline{\psi}_1 \overline{\psi}_2 \rangle \end{pmatrix} = -2\pi \begin{pmatrix} 2\partial_1 & im \\ -im & 2\bar{\partial}_1 \end{pmatrix} \int \frac{\mathrm{d}^2 p}{(2\pi)^2} \frac{e^{i\mathbf{p} \cdot (\mathbf{x}_1 - \mathbf{x}_2)}}{p^2 + m^2} , \qquad (54)$$

where the subscripts indicate the coordinate $\mathbf{x}_i = (x_i, y_i)$ at which the field or derivative is evaluated. Plugging these into the expression for $g_{mm}$, and focusing just on the divergent part, simplifies the result to

$$g_{mm} = \int_0^\Lambda \frac{p \, \mathrm{d}p}{p^2 + m^2} , \qquad (55)$$

where we have introduced an ultraviolet cutoff because the integral is logarithmically divergent. Assuming $\Lambda \ll |m|$, we get

$$g_{mm} \approx \ln\left( \frac{\Lambda}{|m|} \right) . \qquad (56)$$

Renormalization will simply replace the cutoff with a renormalization scale, $\mu$, which, in terms of comparing with $g_{\beta\beta}$ in (48) is just part of the "regular part". As far as the divergence at criticality is concerned, we get the same behavior,

$$g_{mm} \approx \ln \frac{1}{|m|} . \qquad (57)$$

Comparing this result with $g_{\beta\beta}$ in (48), we see that both sides of the mapping retain the salient features, namely the location of the divergence (at $m = 0$) and the type or degree of the divergence (logarithmic).

# 6  Information geometry on state space: coherent fermions

So far, with regards to the 2d Ising model, we have been discussing the Fisher metric on the space of theories parametrized by $J$ and $K$, or $\beta$ (in the isotropic case $J = K = 1$). For the associated free fermion theory there is a single parameter, the fermion mass $m$, which may be obtained as function of $J$ and $K$ when performing the map. However, as we did for the moduli space of scalar field instantons, inspired by the work in [26] on Yang-Mills instantons, we can also consider the information metric on a space of states.

For a general quantum field theory, the quantum analogue of the Fisher metric when working with density matrices is the Bures metric which is defined in the following way [55]. The Bures metric is defined by first defining the Bures distance between two density matrices $\rho_1$ and $\rho_2$,

$$D_B(\rho_1, \rho_2) = 2\Big(1 - \mathrm{tr}\sqrt{\rho_1^{1/2}\rho_2\,\rho_1^{1/2}}\Big)\,,\tag{58}$$

and then expanding this Bures distance to lowest nontrivial order in $d\rho$, where $\rho_2 = \rho_1 + d\rho$. The lowest order is second order, which thus defines a line element and a metric, called the Bures metric. For pure states, the Bures metric reduces to the Fisher metric (up to an overall factor).

We now consider the Bures metric on the space of coherent states, for which it reduces to the Fisher metric. For a single spin, this was calculated in [56]. Here, the spin is parametrized in terms of a normalized three-dimensional vector $(x^1, x^2, x^3) = (\sin\theta\cos\phi, \sin\theta\sin\phi, \cos\theta)$ as

$$|z\rangle = \frac{1}{(1 + |z|^2)^{1/2}}\begin{pmatrix}1\\z\end{pmatrix},\tag{59}$$

where

$$z = \frac{x^1 + ix^2}{1 + x^3} = e^{i\phi}\tan\frac{\theta}{2},\tag{60}$$

and the Fisher metric is found to be that of a two-dimensional sphere,

$$\mathrm{d}s^2 = \frac{\mathrm{d}z\,\mathrm{d}\bar{z}}{(1 + |z|^2)^2} = \frac{1}{4}(\mathrm{d}\theta^2 + \sin^2\theta\,\mathrm{d}\phi^2).\tag{61}$$

An equivalent result is obtained for the Fisher metric of two free Majorana fermions [55]. Denoting theses fermions by $\psi_1$ and $\psi_2$, the coherent state is defined to be

$$|\psi_\lambda\rangle = \frac{1 - \lambda\psi_1^\dagger(k)\,\psi_2(k)}{(1 + |\lambda|^2)^{1/2}}\,|\Omega\rangle\,,\tag{62}$$

where $\lambda$ is a complex parameter and $|\Omega\rangle$ is the unentangled IR state defined by

$$\psi_1(k)\,|\Omega\rangle = 0, \qquad\qquad \psi_2^\dagger\,|\Omega\rangle = 0.\tag{63}$$

For the states (62), the Bures metric is found to be that of a two-dimensional sphere as well,

$$\mathrm{d}s^2 = \frac{\mathrm{d}\lambda\,\mathrm{d}\bar{\lambda}}{(1 + |\lambda|^2)^2}.\tag{64}$$

As an interesting fact we note that according to [33], within information geometry so-called categorical distributions, different from the exponential families discussed in section 2.1 above, lead to spherical Fisher metrics. In information theory, categorical distributions are generalized Bernoulli distributions describing a discrete random variable with more than

two possible outcomes with fixed probability. However, as again discussed in detail in [39], the map between probability distributions and metrics is not bijective, and also other distributions may lead to spherical Fisher metrics. Nevertheless, since bosonic coherent states lead to hyperbolic Fisher metrics [55], it appears as a promising open question to determine how the Pauli principle leads to this different geometric structure for fermions and bosons.

# 7 Different notions of curvature

In this section, we begin by clarifying the various notions of curvature that appear in the literature; in particular, the statement that non-interacting theories are flat properly refers to flatness with respect to the 1-curvature, *not* the 0-curvature leading to a metric connection. The failure to appreciate this distinction seems to have lead to some erroneous and potentially confusing claims. The following will draw primarily from [21]; see also [24] for a brief introduction.

For maximum clarity, let us start be recalling some basic differential-geometric notions. Recall that the covariant derivative $\nabla$ may be expressed in local coordinates as

$$\nabla_X Y = X^i \left( \partial_i Y^k + Y^j \Gamma_{ij}{}^k \right) \partial_k \ , \tag{65}$$

where $X = X^i \partial_i$, $Y = Y^i \partial_i$ are vectors in the tangent space $TS$. If these are basis vectors such that $X^i = Y^i = 1$, then $\nabla_{\partial_i} \partial_j = \Gamma_{ij}{}^k \partial_k$. The vector $Y$ is said to be *parallel* with respect to the connection $\nabla$ if $\nabla Y = 0$, i.e., $\nabla_X Y = 0 \ \ \forall X \in TS$; equivalently, in local coordinates,

$$\partial_i Y^k + Y^j \Gamma_{ij}{}^k = 0 \ . \tag{66}$$

If all basis vectors are parallel with respect to a coordinate system $[\xi^i]$, then the latter is an *affine coordinate system* for $\nabla$. A connection $\nabla$ which admits such an affine parametrization is called *flat*, i.e., the manifold $S$ is flat with respect to $\nabla$.

Now, with respect to a Riemannian metric $g$, one defines

$$\Gamma_{ij,k} = \langle \nabla_{\partial_i} \partial_j, \partial_k \rangle = \Gamma_{ij}{}^l g_{lk} \ , \tag{67}$$

which defines a *symmetric* connection, i.e., $g_{ij} = g_{ji}$. If, in addition, $\nabla$ satisfies

$$Z\langle X, Y \rangle = \langle \nabla_Z X, Y \rangle + \langle X, \nabla_Z Y \rangle \ \ \forall X, Y, Z \in TS \ , \tag{68}$$

or, equivalently,

$$\partial_k g_{ij} = \Gamma_{ki,j} + \Gamma_{kj,i} \ , \tag{69}$$

where $g_{ij} = \langle \partial_i, \partial_j \rangle$, then $\nabla$ is a *metric* connection with respect to the Riemannian metric $g$. Connections which are both metric and symmetric are *Riemannian*.

The above describes the familiar 0-connection, henceforth denoted $\nabla^{(0)}$, with associated connection coefficients $\Gamma^{(0)}_{ij,k}$. The significance of such connections in physics – and Riemannian geometry more generally – is due to the fact that under a metric connection, parallel transport of two vectors preserves the inner product. However, the natural connections on statistical manifolds are generically non-metric, as we shall now explain.

If $S = \{p_\xi\}$ is an $n$-dimensional model as above, we may define the $n^3$ functions $\Gamma^{(\alpha)}_{ij,k}$ which map each point in $\xi$ to

$$\left( \Gamma^{(\alpha)}_{ij,k} \right)_\xi \equiv E_\xi \left[ \left( \partial_i \partial_j \ell_\xi + \frac{1-\alpha}{2} \partial_i \ell_\xi \partial_j \ell_\xi \right) (\partial_k \ell_\xi) \right] \ , \tag{70}$$

where $\alpha \in \mathbb{R}$. This defines an affine connection $\nabla^{(\alpha)}$ on $S$ via

$$\langle \nabla^{(\alpha)}_{\partial_i} \partial_j, \partial_k \rangle = \Gamma^{(\alpha)}_{ij,k} , \tag{71}$$

where $g = \langle \cdot, \cdot \rangle$ is the Fisher metric (4). $\nabla^{(\alpha)}$ is called the $\alpha$-*connection*, and accordingly terms like $\alpha$-flat, $\alpha$-affine, $\alpha$-parallel, etc. denote the corresponding notions with respect to this connection. Note that when $\alpha = 0$ we recover the familiar metric connection above. Indeed, observe that

$$\begin{aligned} \partial_k g_{ij} &= \partial_k E_\xi[\partial_i \ell_\xi \partial_j \ell_\xi] \\ &= E_\xi[(\partial_k \partial_i \ell_\xi)(\partial_j \ell_\xi)] + E_\xi[(\partial_i \ell_\xi)(\partial_k \partial_j \ell_\xi)] + E_\xi[(\partial_i \ell_\xi)(\partial_j \ell_\xi)(\partial_k \ell_\xi)] \\ &= \Gamma^{(0)}_{ki,j} + \Gamma^{(0)}_{kj,i} , \end{aligned} \tag{72}$$

cf. (69). Thus, while $\nabla^{(\alpha)}$ is symmetric for any value of $\alpha$ by definition (cf. (71) and (67)), only the special case $\alpha = 0$ defines a Riemannian connection $\nabla^{(0)}$ with respect to the Fisher metric.

Physically, the significance of the 1-connection lies in the fact that it is intimately related to the Kullback-Leibler divergence or relative entropy. As explained in [21], one can introduce a class of distance-like measures called $\alpha$-divergences, each of which is naturally associated with the corresponding $\alpha$-connection with respect to $g$. For $\alpha = 1$, this is the familiar relative entropy,

$$D(p||q) = \int \mathrm{d}x \, p(x) \ln \frac{p(x)}{q(x)} , \tag{73}$$

where $p, q$ are two points on the manifold $S$, i.e., two different probability distributions. Due to the asymmetry, the divergence is not a true distance metric, but can be seen to be intimately related to the Fisher metric by considering the divergence between infinitesimally separated distributions $p = p(x; \theta)$ and $q = p(x; \theta')$, where

$$p(x; \theta') = p(x; \theta) + \Delta\theta^i \partial_i p(x; \theta) + \dots , \tag{74}$$

where $\Delta\theta^i = (\theta' - \theta)^i$ is some infinitesimal change in the $i^{\text{th}}$ direction. Since the relative entropy is 0 to leading order in this perturbation, the series expansion up to second order reads

$$D\big(p(\theta); p(\theta')\big) = \frac{1}{2} \Delta\theta^i \Delta\theta^j g_{ij} + \dots , \tag{75}$$

where

$$g_{ij} = \partial_i \partial_j D\big(p(\theta); p(\theta')\big) \tag{76}$$

is the Fisher metric introduced above. Thus the curvature induced from the 1-connection is in some sense the natural notion of "statistical curvature" appropriate to the manifold $S$.

Additionally, the 1-connection is intimately associated with the exponential families introduced in section 2.1, namely that the canonical coordinates $[\theta^i]$ provide a 1-affine coordinate system, with respect to which $S$ is 1-flat. To see this, observe that

$$\partial_i \ell(x; \theta) = F_i(x) - \partial_i \psi(\theta) \quad \Longrightarrow \quad \partial_i \partial_j \ell(x; \theta) = -\partial_i \partial_j \psi(\theta) . \tag{77}$$

This implies that

$$\left(\Gamma^{(1)}_{ij,k}\right)_\theta = E_\theta[(\partial_i \partial_j \ell_\theta)(\partial_k \ell_\theta)] = -\partial_i \partial_j \psi(\theta) E_\theta[\partial_k \ell_\theta] = 0 , \tag{78}$$

such that the curvature vanishes identically in this case.

It is also simple to show that for the exponential family $\Gamma_{ij,k}^{(\alpha)}$ is proportional to the third-order moments of $F_i$:

$$\Gamma_{ij,k}^{(\alpha)} = \frac{1-\alpha}{2} \langle (F_i - \langle F_i \rangle)(F_j - \langle F_j \rangle)(F_k - \langle F_k \rangle) \rangle , \tag{79}$$

which indeed vanishes identically for $\alpha = 1$.

As mentioned above, non-interacting theories are only flat in the sense of 1-flatness, cf. the Gaussian example in section 2.2. But this same flatness holds for any model that can be put in the form of an exponential family, including the Ising model on theory space spanned by its couplings that we discussed above. Furthermore, we found that for the non-interacting Gaussian theory, the usual 0-curvature is a negative constant, while for the Ising model, it changes depending on the values of the couplings. It remains an open question as to precisely what information about the underlying physical theory is encoded in the different curvatures. For example, the divergence in the 0-curvature of the Ising model correctly captures the phase transition; however, note that the 1-curvature remains zero even along this critical line, and is therefore completely insensitive to the critical behaviour. An important question for the future is thus to determine which physical behaviour is captured by the different curvatures.

As a noteworthy fact in view of establishing new field theory/gravity dualities, we point out that for non-metric curvatures, in 2+1 dimensions a gravity action may be obtained from the Chern-Simons action

$$\mathcal{S} = \frac{1}{4\pi} \operatorname{tr} \int \left( \Gamma \wedge \mathrm{d}\Gamma + \frac{2}{3} \Gamma \wedge \Gamma \wedge \Gamma \right) , \tag{80}$$

for which the equation of motion implies that the covariant derivative of the curvature vanishes,

$$\mathrm{d}\Gamma + \Gamma \wedge \Gamma = 0 . \tag{81}$$

Obviously, the case (78) in which the 1-curvature vanishes itself is a special solution to the more general equation of motion (81). This suggests a possible duality between field theories leading to an exponential family and gravity actions involving non-metric curvatures.

## 8  Discussion

In this paper, we have collected and discussed a number of general lessons that we feel are important in the application of information geometry to quantum field theory and to the AdS/CFT correspondence. Our discussion was framed around some simple examples: exponential families of probability distributions (of which the Gaussian is a representative member), scalar field instantons, and the 2d classical Ising model on a square lattice and its mapping to the theory of free massive Majorana fermions. For clarity, we enumerate these general lessons here:

1. *Infinitely many different probability distributions give the same Fisher metric. The Fisher metric inherits the symmetries of the probability distribution, but the probability distribution does not necessarily need to enjoy all the symmetries of the Fisher metric.*

   In many of the cases studied in the literature, the probability distribution enjoys precisely the translation and scaling symmetries that suffice to force the Fisher metric

to be AdS. In the light of our investigations, it is conceivable that there are other dualities relating quantum field theories to geometries. However, the point raised above has to be taken into account when studying these, and of course the most relevant question is what determines the dynamics of the dual gravity theory.

2. *There are two basic ways of applying information geometry to quantum field theories: one can compute a metric on the space of theories parametrized by coupling constants or on the space of states of a given theory with a fixed set of coupling constants.*

   For example, saying that the Fisher metric on a free real massive scalar field is $AdS_2$ is ambiguous and potentially misleading. This happens to be the Fisher metric on a particular set of coherent states parametrized by one complex coefficient [55], but is unrelated to the Fisher metric on the space of such theories parametrized by the mass. Which prescription one uses depends on what one wishes to study, and it is important to keep the distinction in mind.

3. *The Fisher metric on a set of states of a quantum field theory is sensitive to whether or not the theory has a stable potential.*

   We demonstrated this phenomenon with the example of a massless real scalar field in four Euclidean dimensions with an inverted $\phi^4$ potential. We considered the moduli space of instantons for this theory, parametrized by the center and width of the instanton. Symmetry arguments along the lines of point 1 above imply that the Fisher metric ought to be $AdS_5$, and this can indeed be arranged to be the real part of the metric. However, the Fisher metric is necessarily complex in this case and thus cannot be considered real Euclidean $AdS_5$.

4. *There are many different connections one can define in information geometry, most of which are not metric compatible with respect to the Fisher metric.*

   This technical distinction is obscured in some of the existing literature, both in recent physics works and older works in statistics while the basics of information geometry were still under development. It is relevant in light of claims that free theories lead to flat geometries, and a finer appreciation of these various curvature notions may be important for determining precisely what information about the underlying physical theory is encoded in the geometry.

Finally, let us remark on a couple interesting and potentially fruitful connections between information geometry and holography, namely holographic RG and complexity:

**Holographic RG**

Intuitively, the flow along the RG can be thought of as a coarse-graining of degrees of freedom from the UV to the IR. Consequently, if one considers two nearby theories in the UV, more and more measurements will be necessary to distinguish them as one flows to the IR, cf. the intuition below (20). That is, the inability to probe fine-grained correlators implies a loss of distinguishability between nearby theories. In [44], this idea was made more precise by calculating the distance between quantum field theories using the Zamolodchikov metric, which is proportional to the Fisher metric studied here. In the context of the emergent spacetime or "it from qubit" paradigm, in which one takes the boundary CFT as ontologically prior and attempts to derive the bulk AdS along with its dynamics, this line of reasoning suggests that the classical spacetime deep in the IR may result from a coarse-graining procedure over suitably-parametrized UV theories. A related observation was made in [36] where it was suggested that the expectation value

in the Fisher metric may be thought of as a statistical average over quantum fluctuations that gives rise to the classical spacetime. Thus, despite the cautionary lessons above, we regard it as a very interesting open question as to whether the connection between the information content of the field theory and the geometry resulting from the Fisher metric (or its quantum analogues) can be utilized to shed further light on gauge/gravity duality, and perhaps lead to "information/geometry" dualities in a wider context.

**Complexity**

Another potential application is in generalizing holographic complexity [46] to interacting and ultimately strongly coupled theories. While a great deal of exciting progress has been made in free theories (see [57–64] and related work), and attempts have been made to go beyond this restriction (see in particular [65], as well as [66–69]), a satisfying prescription for defining and working with complexity in holographic CFTs remains elusive. However, the basic idea underlying existing approaches is to geometrize the problem, and define complexity in terms of the distance between quantum states. Insofar as information geometry already provides what is in some sense the intrinsic geometry for a given distribution, it is therefore natural to ask whether this framework can be used to define complexity in general theories, including holographic CFTs.

In principle at least, one can already do this for any of the models considered above. For example, we could use the results of section 5 to define complexity for the 2d Ising model as the minimum geodesic distance between theories with different couplings. Practically however, the metric is so unwieldy that we could only proceed with our curvature calculation numerically, and even an approximate analytical expression for the geodesics seems beyond reach. In the case of AdS/CFT, one again encounters the question of how to suitably parametrize the boundary theory in order to apply this framework, which may largely determine the physical meaning of the results. One would also need to contend with the first lesson in the list above, namely that very different distributions – and hence, states/theories – may yield the same geometry, and hence the same complexity. Whether this is because the Fisher metric is not a sufficiently refined means of probing these theories, or hints at some deep connection between them, remains to be seen. Nonetheless, in light of the significant efforts to quantify complexity seen in the past couple years, it seems worth investigating whether methods from information geometry can be fruitfully applied to go beyond the limits of current approaches.

# 9   Acknowledgements

We are grateful to Jan de Boer for discussions and hospitality at the University of Amsterdam. We also thank Souvik Banerjee and René Meyer for discussions. K. G. acknowledges funding through a Hallwachs-Röntgen fellowship. J. E. and K. G. are supported by the Würzburg-Dresden Cluster of Excellence on Complexity and Topology in Quantum Matter—ct.qmat (EXC 2147, Project-id No. 39085490). R. J. is a member of the Gravity, Quantum Fields and Information group at AEI, which is generously supported by the Alexander von Humboldt Foundation and the Federal Ministry for Education and Research through the Sofja Kovalevskaja Award.

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
