# Peer review of "Information geometry in quantum field theory: lessons from simple examples"

_SciPost Physics_

## Round 1 · Referee Report · Anonymous (Referee 1) · 2020-3-8

Strengths

1) A good overall discussion of information geometry within quantum field theories.
2) Clearly presented calculations.
3) Clarifications of common misunderstandings.

Weaknesses

1) Not highly novel
2) Somewhat speculative

Report

In this paper, Erdmenger et al. give an overview of the applications of information geometry within quantum field theories. After providing a clear explanation of the statistical origins of the Fisher Metric they discuss the exponential family of distributions and show how they can give rise to hyperbolic information manifolds. They show how symmetries of the distribution give rise to symmetries of the metric but that this is not always the case.

They then explain how instabilities in a field theory can show up in the Fisher metric as complex components. This is a novel application of information geometry in this context. There seem to be a number of issues with this calculation however. The Hitchin procedure takes the Lagrangian and defines it as the probability density. However, in this example such a direct mapping is not clear as the Lagrangian is not positive-definite on the solution of the instanton. This needs clarification and I believe that a more general proof, or at least another example of an instability showing up in this relationship between instability and complexity is needed.

The next two sections are split into studying the information geometry of the theory space and of the state space respectively. In theory space they show how the Ising model information metric encodes aspects of the phase structure of the theory, and then that parts of this are also encoded in the dual one dimensional fermionic theory. In state space, the Bures metric is used to study the information space of coherent fermions showing how a spherical metric can come about.

In the last main section the fact that notions of curvature differ between the physics and information theory community is explained in detail.

Finally the authors summarise and conclude with some speculations about the use of information geometry in holographic RG and complexity within a gauge/gravity context.

Overall this paper gives a good overview of the potential uses of information geometry in quantum field theories with some interesting examples and some clarifications of common misconceptions. I believe that the paper could be strengthened with more general statements about the correspondence, particularly between unstable configurations and the complexity of the metric.

Requested changes

1) Page 5, below equation 2. The n in the size of the space of variables and that in the parameter space should be different
2) Page 7, below equation 20. This is not AdS as AdS is Lorentzian and this is not. This is a hyperbolic metric.
3) Page 7, "yields and" -> "yields an"
4) The terms Fisher matrix and Fisher metric are used interchangeably. This should be made uniform.
5) Page 9. It is not clear where the a comes from in equation 29, nor where the zeta goes. This should be clarified.
6) Page 9. That the Lagrangian density in this case isn't a probability density needs to be made clear.
7) Page 9. A more general statement about the relationship between complexity and instability needs to be made.
8) Page 10. "in thermodynamic"->"in the thermodynamic".
9) Page 10, equation 36. k should be replaced with another letter as this equation contains k and K.
10) Page 13, Figure 1. It's not clear why the imaginary part line needs to be there when it's clearly zero from the equation.
11) Page 14, "the one can"->"one can".
12) Page 15, equation 58. The Bures distance is not correct here.
13) I believe that section 7 should come earlier in the text as it does not follow on naturally from section 6.

  • validity: high
  • significance: ok
  • originality: ok
  • clarity: high
  • formatting: excellent
  • grammar: perfect

Author:  Ro Jefferson  on 2020-04-06  [id 791]

(in reply to Report 1 on 2020-03-08)

We wish to thank the referee for their many helpful comments regarding our paper. Here we wish to describe the changes that we have made to our article (version 2) in response to the referee's points.

First, let us respond to the comment regarding the instabilities. We are grateful to the referee for raising this point. Indeed, we agree that the Lagrangian density, as we have written it and as it is customarily written, is not positive semi-definite when evaluated on the instanton solution. Therefore, this does not give a well-defined probability distribution or Fisher metric. We state this explicitly in the revised version of the text. Additionally, we point out that we can add a total derivative to the Lagrangian density that does not contribute any boundary terms. This does not alter the physical system in any way, of course. However, we can do this in such a way as to produce a Lagrangian density which is positive semi-definite when evaluated on the instanton solutions and therefore does give a well-defined probability distribution. The resulting Fisher metric is in fact hyperbolic space.

Second, to increase the novelty of the paper, we have added a new section 7 which deals with the Bures metric on the space of coherent states of one free complex bosonic scalar field. Previously, we indicated that it would be an interesting problem to trace precisely how the fermionic versus bosonic nature of the field theory yielded a sphere versus a hyperbolic metric, respectively. We work this out explicitly by demonstrating that the density matrix has an SO(3) symmetry group in the fermionic case and an SO(2,1) symmetry group in the bosonic case.

With respect to the “Requested changes” in particular:

  1. Indeed, the variable 'n' in the size of the space of variables and that in the parameter space should be different in general. We have changed the one for 'X' from 'n' to 'm'.

  2. We have included the following qualifying statement: "Note that the Fisher metric is always of Euclidean signature, or possibly degenerate, since the Fisher metric is positive semi-definite. To get a metric of Lorentzian or any other mixed signature requires Wick rotation a posteriori. We will henceforth take this for granted and drop the 'Euclidean' qualifier."

  3. We have fixed this typo.

  4. We have changed “matrix” to “metric” wherever appropriate to ensure uniform language.

  5. The constant 'a' was just an overall coefficient in the Fisher metric which we introduced by hand, since an overall coefficient is immaterial anyway. In any case, in response to this comment, we simply removed 'a' altogether. The factor of 'ζ' drops out of the integrals in the Fisher metric because of translation-invariance. This is shown explicitly by the change of variables which is now in equation (32).

  6. This statement is now made explicit immediately following the expression for the Lagrangian density evaluated on the instanton solution, which is now equation (35).

  7. We are unaware of a general relationship between complexity and instability. However, in response to this comment, we now remark that it is possible to integrate the Lagrangian density by parts to get a different Lagrangian density which does describe a well-defined probability distribution and which, in this example, yields the hyperbolic Fisher metric. This is a general lesson one has to keep in mind when using the Hitchin proposal since adding total derivatives to Lagrangian densities that contribute no nontrivial boundary terms is not supposed to actually change the system. Nevertheless, this can make the difference between having and not having a well-defined probability distribution in the first place. This is reflected in appropriate changes to the introduction and conclusion.

  8. We have fixed this typo.

  9. We have change 'k' to 'κ' (kappa).

  10. We have removed the unnecessary imaginary part line and explicit reference to it in the description of the Figure 1.

  11. We have fixed this typo.

  12. We are using the Bures distance as defined in reference [9] of the article. We are aware of the difference relative to the literature and have commented on it in Footnote 7 to avoid any confusion.

  13. The previous section 7 (now section 8) is about different notions of curvature. It is really an independent section and does not thematically follow strongly from the previous sections of the paper, and we felt that the technical detail might furthermore detract from the main thread if placed in earlier sections. It also harkens back to the statement about free theories having flat Fisher metrics from earlier on in the paper. This is why we prefer to leave this section at the end of the paper. We have however slightly reorganized the discussion of divergences to improve the flow of the paper (some material is now mentioned in section 6), and edited the discussion about the significance of the 1-curvature.

In addition to the above, we have made a number of further changes which we feel improves the article:

  1. We added a useful equivalent expression for the Fisher metric in equation (8).

  2. As already mentioned in Point 13 above, we have included an expanded discussion of divergence functions in Section 6. The point of this discussion is to motivate the Bures metric in the first place. We state that, when talking about quantum states, including mixed states, we can simply replace real-valued probability distributions in the divergence functions with density matrices.

  3. We have also included a new appendix where we demonstrate that the Fisher metric is equal to the second-order expansion of the α-divergence for any α and that the α-connection is derived from third-order expansion.

  4. We have changed the notation of the coherent state for the two Majorana fermion theory in what is now equation (79). Previously, this was written in terms of two field operators ψ(k) and ψ(k). This did not make the “coherent” nature of the state obvious. It should be obvious now since it is written in terms of the standard exponential form and the fermionic nature is made explicit with the anticommutation relations in equation (73).

  5. As previously mentioned, we added a new section 7 about the bosonic coherent state and how the Bures metrics for the fermionic and bosonic coherent states reflect the SO(3) and SO(2,1) symmetries of the density matrix, respectively. This is reflected by corresponding additions to the introduction and conclusion.

  6. We have corrected a typo (previously, g_{ij} was referred to when mentioning symmetric connections below what is now eq. (89), rather than Γ_{ij,k}), made a few minor language alterations for clarity, and added a couple references.

  7. In the paragraph between equations (86) and (87), which are now (97) and (98), we have removed the sentence “Furthermore, we found that for the non-interacting Gaussian theory, the usual 0-curvature is a negative constant, while for the Ising model, it changes depending on the values of the couplings”. It is not true that non-interacting Gaussian theories have negative constant 0-curvature. We also have no need for this statement anyway, so we simply remove it.

  8. After point 4 of section 9, we inserted a paragraph suggesting a possible application to the 3d Ising model and its connection with string theory proposed by Polyakov and studied recently Iqbal and McGreevy (reference [61] in the article).

Anonymous on 2020-05-04  [id 815]

(in reply to Ro Jefferson on 2020-04-06 [id 791])

I believe that these corrections and changes sufficiently resolve the original issues with the article and it is now an excellent addition to the advancement of this particular topic.

---

## Editorial Decision

resubmitted